# Non-Fermi Liquid Behavior in the Three-Dimensional Hubbard Model

**Samuel Kellar** [1]**, Ka-Ming Tam** [1,2,]***and Juana Moreno** [1,2,]*

1  Department of Physics and Astronomy, Louisiana State University, Baton Rouge, LA 70803, USA
2  Center for Computation & Technology, Louisiana State University, Baton Rouge, LA 70803, USA
*  Correspondence: phy.kaming@gmail.com (K.-M.T.); moreno@lsu.edu (J.M.)

**Abstract:** We present a numerical study on the non-Fermi liquid behavior of a three-dimensional strongly correlated system. The Hubbard model in a simple cubic lattice is simulated by the dynamical cluster approximation; in particular, the quasi-particle weight is calculated at finite dopings for a range of temperatures. By fitting the quasi-particle weight to the marginal Fermi liquid form at finite doping near the putative quantum critical point, we find evidence of a separatrix between Fermi liquid and non-Fermi liquid regions. Our results suggest that a marginal Fermi liquid and possibly a quantum critical point exist in the non-symmetry broken solution of the three-dimensional interacting electron systems. We also calculate the spectral function, close to the half-filling, and we obtain evidence of pseudogap.

**Keywords:** metal insulator transition; Mott transition; random disorder; dynamical cluster approximation; non-Fermi liquid; marginal Fermi liquid; Hubbard model





## 1. Introduction

The Fermi liquid theory was an important milestone in the field of condensed matter physics [1–3]. This theory encapsulates almost all the properties of metallic interacting Fermionic systems. The fundamental assumption is that the interacting system can be described by adiabatically turning on the interaction. All the quantum numbers of the non-interacting system remain intact; specifically, the momentum remains a good quantum number for characterizing the excitations in metallic systems.

There are notable exceptions to the Fermi liquid model, which have been discovered over time. The most prominent is one-dimensional systems, in which different quantum numbers emerge due to the spin-charge separation [4]. However, in general, it is rather difficult to violate the assumptions of the Fermi liquid theory as dictated by the phase space restriction in the particle-hole diagrams [5,6]. The general premise on systems violating the Fermi liquid theory is that either its density of states or its intrinsic interactions become singular resulting in a strong correction to the Fermi liquid predictions.

Along this line of thought, a possible cause of non-Fermi liquid behavior is the proximity to a quantum critical point. Since a quantum critical point leads to long wavelength fluctuations [7–10], the effective coupling of the electrons due to such fluctuations might become singular. However, it is usually difficult to obtain clean experimental evidence for a quantum critical point. In particular, the metallic critical point studied in this paper might be difficult to observe since it may be preempted by other orderings, such as superconducting pairing [11]. Remarkable experimental progress has been made recently on magnetic quantum critical points on Kondo lattice materials [12,13].

We note that for the two-dimensional lattice, another possible route to break away from the Fermi liquid is the existence of a van Hove singularity [14–19]. The simple phase space argument for the Fermi liquid is invalidated by the divergent density of state at the Fermi level.

A prominent example of non-Fermi liquid behavior is the high-temperature superconducting cuprates [20], which display non-Fermi liquid behavior immediately above the superconducting dome. Cuprates have been extensively studied over the past three decades, as it is widely believed that a key to understanding their high-temperature superconductivity is to comprehend the non-Fermi liquid, often denoted as strange metal [21–26]. A theory that captures a lot of the properties of the strange metal is the theory of the marginal Fermi liquid [22,23,27,28]. The key idea of the marginal Fermi liquid theory is that the quasi-particle damping is much increased to the point that quasi-particle excitations cannot be properly defined or normalized.

For the Fermi liquid, the quasi-particle damping reflects the Lorentzian shape of the excitation spectral weight. The imaginary part of the Fermi liquid self-energy scales as the square of the energy, $\Im[\Sigma(\omega)] \sim \omega^2$, although strictly speaking, a logarithmic correction is acquired at two dimensions [29]. For the three- or higher-dimensional case, it can be shown that the Fermi liquid is stable against particle-hole excitations either using perturbation theory or a modern functional renormalization group [5,6]. For the marginal Fermi liquid, the self-energy scales linearly with respect to the energy, $\Im[\Sigma(\omega)] \sim \omega$. This is the borderline case for which the coherent excitations can be defined. The marginal Fermi liquid theory naturally explains one of the most interesting characteristics of the strange metal—its linear resistivity.

Numerically studying the non-Fermi liquid is challenging as it involves interacting Fermions at finite doping. The Dynamical Cluster Approximation (DCA) has been used to demonstrate the existence of a quantum critical point and its marginal Fermi liquid behavior in the two-dimensional Hubbard model [30]. This work extends the prior study to the three-dimensional system. This paper is organized as follows. In Section 2, we describe the model and the method and review previous DCA studies of non-Fermi liquid and quantum criticality of the Hubbard model in two dimensions. The results of the quasi-particle weight and the estimate of the crossover temperature between Fermi liquid and marginal Fermi liquid are described in Section 3. We then conclude and discuss possible future work on providing additional evidence corroborating the existence of a quantum critical point in the three-dimensional Hubbard model. A short description of the DCA and a discussion of the minus sign problem in the QMC impurity solver are presented in Appendix A.

## 2. Model and Methods

### 2.1. Model

Our starting point is the Hubbard model

$$H = -t \sum_{<i,j>,\sigma} c_{i\sigma}^{\dagger} c_{j\sigma} + U \sum_i n_{i\uparrow} n_{i\downarrow} - \mu \sum_{i,\sigma} n_{i,\sigma}, \tag{1}$$

where $c_{i,\sigma}^{\dagger}$ and $c_{i,\sigma}$ are the creation and annihilation operators for electrons at site $i$ with spin $\sigma$, $n_{i,\sigma}$ is the number operator for site $i$ and spin $\sigma$, $t$ is the hopping energy between nearest neighbors in a simple cubic lattice, $U$ is the on-site repulsive coupling, and the chemical potential, $\mu$, sets the filling of the system. The hopping, $t$, is set to 0.25, and it is used as the energy scale. The bare bandwidth is $W = 12t = 3$. The on-site interaction is set to $U = 0.75$ W.

The three-dimensional Hubbard model acts as a bridge between the dynamical mean field theory (DMFT), exact at infinite dimensions, and the two-dimensional Hubbard model used to model the cuprate superconductors [31]. The three-dimensional Hubbard model has primarily been studied in terms of its antiferromagnetic properties at half-filling [32–35]. Recently, the doped antiferromagnetic critical point has also been studied by the dynamical vertex approximation [36,37]. This research intends to further explore the features of the metallic phase of a doped three-dimensional Hubbard model.

We use the Dynamical Cluster Approximation (DCA) to solve the Hubbard model. The DCA employs clusters on a periodic lattice embedded in a dynamical mean field [38,39].

These clusters allow the incorporation of non-local corrections to the DMFT. The size of the cluster controls which non-local corrections are included. As the cluster size increases by including more impurity sites, $N_c$, non-local corrections at larger distances are captured. If the DCA includes only one site, $N_c = 1$, it is equivalent to the DMFT. The mapping of the lattice model to a finite cluster is accomplished by coarse-graining the lattice problem in reciprocal space. The DMFT and other local approximations such as the CPA are equivalent to neglecting momentum conservation at all internal vertices of the self-energy so that the problem is coarse-grained over the entire Brillouin zone. The DCA systematically restores momentum conservation by introducing a coarse-graining scale $\Delta k$. As a result, non-local dynamical correlations of range $\sim \pi / \Delta k$ are treated accurately, since the lattice problem is mapped onto a periodic cluster of roughly this size.

The cubic lattice may be tiled by cubic clusters when the number of impurity sizes is a perfect cube. That largely limits the choices available. In order to find intermediate sizes, a Betts lattice is employed [40]. We choose a sixteen-site cluster to keep the computational time accessible while still including a reasonable number of cluster points in the first Brillouin zone.

### 2.2. Dynamical Cluster Approximation Cluster

The DCA approximation partially restores the spatial resolution missed in the DMFT by dividing the first Brillouin zone into a finite number of boxes. Each box has identical shape and size, and the entire first Brillouin zone is tiled by these boxes. The boxes do not overlap with each other. For the three-dimensional simple cubic lattice, we tile the entire first Brillouin zone with boxes with the shape of a parallelepiped. The size and shape of the parallelepiped can be defined by the following three edge vectors:

$$\mathbf{a}_1 = (\frac{\pi}{2}, \frac{\pi}{2}, 0) \quad \mathbf{a}_2 = (\frac{\pi}{2}, -\frac{\pi}{2}, -\frac{\pi}{2}) \quad \mathbf{a}_3 = (-\frac{\pi}{2}, \frac{\pi}{2}, -\frac{\pi}{2}).$$

The volume of each parallelepiped is $|(\mathbf{a}_1 \times \mathbf{a}_2) \cdot \mathbf{a}_3| = \pi^3/2$. The shape of a parallelepiped is depicted in Figure 1. The first Brillouin zone is tiled by 16 of these parallelepipeds. The center of each parallelepiped defines a momentum point, and we denote it as $\mathbf{K}_i$. Due to the cubic lattice symmetries, the resulting 16 momentum points are not all unique. The location of the 8 unique points in the first Brillouin zone is

$$(-\frac{\pi}{2}, -\frac{\pi}{2}, \pi), \quad (-\frac{\pi}{2}, -\frac{\pi}{2}, 0), \quad (0, 0, \pi), \quad (-\frac{\pi}{2}, \frac{\pi}{2}, \frac{\pi}{2}),$$
$$(0, 0, 0), \quad (0, \pi, \frac{\pi}{2}), \quad (\pi, \pi, \pi), \quad (\pi, \pi, 0).$$

The periodic boundary condition will be respected by the tiling as any point outside the first Brillouin zone will be mapped back inside at an equivalent point.

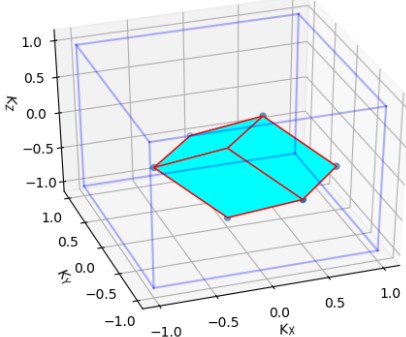

**Figure 1.** The parallelepiped in the figure represents one of the 16 boxes that tiled the entire first Brillouin zone. The outline of the cube delineates the boundary of the first Brillouin zone. $K_x$, $K_y$ and $K_z$ are in units of $\pi$.

### 2.3. Continuous Time Quantum Monte Carlo and Simulation Parameters

We solve the model by the DCA; specifically, we calculate the Green's function. The basic idea of the DCA is to map the interacting lattice model into an impurity cluster model with the hybridization between the impurity cluster and electrons bath determined self-consistently. We outline the approximation and limitations of the DCA in Appendix A. For a more in-depth discussion, we refer the readers to the review paper and the references therein [39].

The effective impurity cluster problem is then solved by numerical methods. In this paper, we employ the continuous time quantum Monte Carlo (CTQMC) to solve the impurity cluster problem [41–44]. For an unbiased Monte Carlo sampling, the measurements should be uncorrelated. For this to occur, the samples must be sufficiently different. In order to minimize the correlations between measurements after the system was warmed up, the majority of proposed changes were flips of the vertices spins [44]. This occurred with a probability of about 70%, while additions and removals of vertices both occurred at a rate of about 15%. The DCA is employed on a high-performance supercomputer. Monte Carlo measurements in the order of $10^6$ were performed with 100 proposed Monte Carlo steps between each measurement for each iteration of the DCA self-consistent cycle.

### 2.4. Locating the Fermi Surface

Our DCA calculation provides the self-energy at 16 points ($\mathbf{K}_i$) in the first Brillouin zone and for four hundred Matsubara frequency points. In order to find the quasi-particle weight, we identify the Fermi surface along the $< 1, 1, 1 >$ direction by calculating the $\max(|\nabla n(\mathbf{k})|)$, where $n(\mathbf{k})$ is the occupation number. Once the momentum of the Fermi surface in the chosen direction is identified, we calculate the quasi-particle weight at that momentum point.

In order to achieve a sufficient resolution to identify the Fermi surface, we use an interpolation method. Since it is important to include information from all points due to the limited resolution in three dimensions with only 16 points, we use an inverse distance weighting scheme [45]. This interpolation method weights the closest data points with a larger value than the further points. The interpolation follows the formula

$$u(x) = \begin{cases} \frac{\sum_i w_i(x) u_i}{\sum_i w_i(x)}, & d(x, x_i) \neq 0 \text{ for all } i \\ u_i, & d(x_i, x) = 0 \text{ for some } i, \end{cases} \tag{2}$$

where $i$ sums over all cluster points, and $u_i$ are the known values. The location being interpolated is labeled $x$, the locations of the known values are $x_i$, $d(x, x_i)$ is the distance of the interpolated point to the known point, and $w_i$ is the weighting function given to each point. The weighting function is defined by

$$w_i(x) = \frac{1}{d(x, x_i)^p} \tag{3}$$

where $p$ is a parameter chosen to control the rate with which the weight drops off over a distance. In this research, a value of 6 was found to be ideal for the parameter $p$.

### 2.5. Quasi-Particle Weight

The lowest Matsubara frequency point of the self-energy at the Fermi surface is then used to calculate the quasi-particle weight. Quasi-particle weight is defined in terms of the real frequency self-energy. In order to relate the quasi-particle weight to the Matsubara self-energy, the following process is followed. The quasi-particle weight is related to the retarded self-energy by,

$$Z_{\mathbf{k}} = \frac{1}{1 - \partial_\omega \Re[\Sigma(\mathbf{k}, \omega)]|_{\omega=0}}. \tag{4}$$

Analytic continuation on the numerical data can be bypassed by taking the derivative of the Kramers–Kronig relation and then using the analytic continuation of the self-energy [46]. The lowest Matsubara frequency point ($\omega_0 = \pi T$) at the Fermi surface is then used to calculate the quasi-particle weight from the imaginary part of the self-energy in Matsubara frequency,

$$Z_{\mathbf{k}} \approx \frac{1}{1 - \Im[\Sigma(\mathbf{k}, i\omega_0)]/\omega_0}.$$ (5)

*2.6. Fitting of the Quasi-Particle Weight*

Our approach for distinguishing the marginal Fermi liquid from the Fermi liquid is based on the frequency dependence in the quasi-particle weight. We first consider the imaginary part of the real frequency self-energy of both the Fermi liquid and the marginal Fermi liquid [6,29,30]. For the Fermi liquid, the imaginary part of the self-energy has the form

$$\Im[\Sigma_{FL}(\omega)] = -\alpha \max\left(\omega^2, T^2\right),$$ (6)

where $\alpha$ is a positive constant.

On the other hand, the imaginary part of the marginal Fermi liquid self-energy has the form

$$\Im[\Sigma_{MFL}(\omega)] = -\alpha \max(|\omega|, T).$$ (7)

Near the putative quantum critical point, at temperature $T_X$ and frequency $\omega_X$, the single-particle properties of the model are observed to cross over from Fermi liquid to marginal Fermi liquid. Vidhyadhiraja et al. [30] proposed to write down the self-energy in terms of the so-called crossover form

$$\Im[\Sigma_X(\omega)] = \begin{cases} -\alpha\omega_X \max(|\omega|, T) & \text{for } |\omega| > \omega_X \text{ or } T > T_X \\ -\alpha \max\left(\omega^2, T^2\right) & \text{for } |\omega| < \omega_X \text{ and } T < T_X \end{cases},$$ (8)

and using analytic continuation with a frequency cutoff $\omega_c$ of the order of the bandwidth, $\Sigma(\mathbf{k}, i\omega_n) = -\int_{-\omega_c}^{\omega_c} \frac{d\omega \Im[\Sigma(\mathbf{k}, \omega)]}{\pi(i\omega_n - \omega)}$. They proposed a crossover form for the self-energy between the marginal Fermi liquid and the Fermi liquid states [30,47],

$$\left(-\frac{\pi}{2\alpha}\right)\frac{\Im[\Sigma(i\omega_0)]}{\omega_0} = T\Theta(T_X - T)\left[\frac{\omega_X}{T} + 0.066235\right.$$
$$\left. - (0.308\frac{\omega_X}{\pi T} + \pi \tan^{-1}\frac{\omega_X}{\pi T}) - \frac{\omega_X}{2T}\ln\left(\frac{\omega_X^2 + \pi^2 T^2}{(1 + \pi^2)T^2}\right)\right]$$
$$+ \omega_X\left[0.0981 + \frac{1}{2}ln\left(\frac{\omega_c^2 + \pi^2 T^2}{(1 + \pi^2)T^2}\right)\right]. \quad (9)$$

The fitting parameters for this form are the scaling parameter $\alpha$, the crossover temperature $T_X$, the crossover frequency $\omega_X$, and the cutoff energy $\omega_c$.

## 3. Results

For the two-dimensional Hubbard model, the DCA calculations suggested there is a quantum critical point at a critical value of doping [30]. Right above the critical doping value, the system is in a pseudogap phase. Below the critical doping value, the system is the Fermi liquid. Marginal Fermi liquid is located around the critical doping bounded by the above two crossover lines. Our major goal is to present numerical evidence to demonstrate that the phase diagram of the three-dimensional case has a strong resemblance to that of the two-dimensional case. The schematic phase diagram is shown in Figure 2.

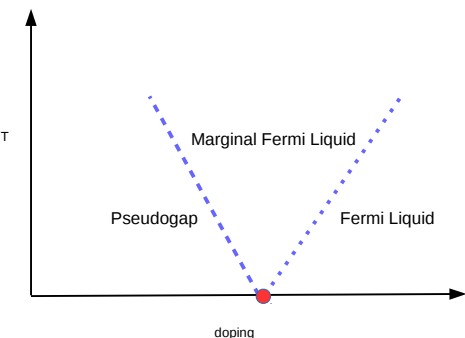

**Figure 2.** Schematic diagram of the putative phases of the paramagnetic solution of the three-dimensional Hubbard model. The vertical axis is temperature and the horizontal axis is doping level. The red dot represents the quantum critical point. The dashed line represents a crossover between the pseudogap and the marginal Fermi liquid. The dotted line represents a crossover between the Fermi liquid and the marginal Fermi liquid. We have not determined the crossover line between the pseudogap phase and the marginal Fermi liquid in this study. The location of the quantum critical point for the parameters of the Hubbard model that we used ($U = 9t$) is estimated to be around filling, $N = 0.95$. The crossover line between the Fermi liquid and the marginal Fermi liquid is determined by the quasi-particle weight. See details in Section 2.6. The precise shape of the crossover lines is dictated by the dynamical exponent of the quantum critical point, we have not attempted to estimate its value in the present study.

*3.1. Quasi-Particle Weight*

In a Fermi liquid, the quasi-particle weight has a finite value at zero temperature. While the $T = 0$ limit cannot be simulated by the quantum Monte Carlo solver for the DCA, the saturation of the quasi-particle weight at low temperatures allows an extrapolation to a finite value at $T = 0$.

For low dopings at low temperatures, the systems should harbor either a Mott insulating state or pseudogap phase. The quasi-particle weight of both the Mott insulator and pseudogap phase has no residual value at zero temperature as the density of state at the Fermi level is zero. Then, the low-temperature simulation will show a stark contrast to that of a Fermi liquid and will be readily distinguishable due to the rapid decrease in quasi-particle weight as temperature decreases.

The results included in Figure 3 show a clear separation at a finite filling between Fermi liquid and non-Fermi liquid behavior. We have run simulations up to a temperature of $T = 0.0125$, below which the computational cost becomes prohibitively expensive. Above the filling $N = 0.95$, the quasi-particle weight tends towards zero as the temperature decreases. The closer to half-filling, the faster the quasi-particle weight asymptotically approaches zero. This can be interpreted as the formation of a pseudogap region in which the spectral weight around the Fermi level is reduced, and we corroborate this interpretation by calculating the spectral function as shown below.

Around the filling factor of $N = 0.9$, the quasi-article weight goes asymptotically towards a finite value at low temperature. This indicates the pseudogap phase ceases to exist at lower fillings. We again corroborate this interpretation by calculating the spectral function at $N = 0.9$, and there is no evidence of the formation of a spectral gap near the Fermi level.

In addition, at high temperatures, the system shows some evidence of being in a marginal Fermi liquid state. Using the method as discussed in Section 2.6, we fit the quasi-particle weight for the fillings $N = 0.9, 0.85, 0.8$ and $0.75$ into the crossover form (Equation (9)). The fit of the quasi-particle weight to the crossover form is shown as dotted lines in Figure 3.

From the fit to Equation (9), we extract the crossover temperature, $T_X$, displayed in Figure 4. The results show that the crossover temperature monotonically decreases as the doping decreases. This indicates a possible zero temperature crossover at a finite doping in agreement with the finding that the non-Fermi liquid state begins around $N = 0.95$, as shown in Figure 3. This can be interpreted as evidence of the existence of a quantum critical point at a finite doping [30].

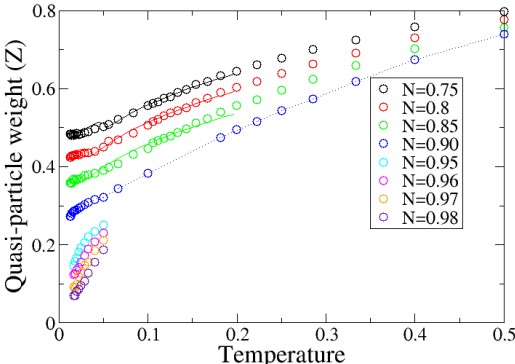

**Figure 3.** The quasi-particle weight at the Fermi surface along the $< 111 >$ direction as a function of filling, $N$. There is a clear separatrix between the behavior in highly doped systems and those near half-filling. The division occurs at about $N = 0.95$. The dotted lines show the crossover form fit (see Equation (9)).

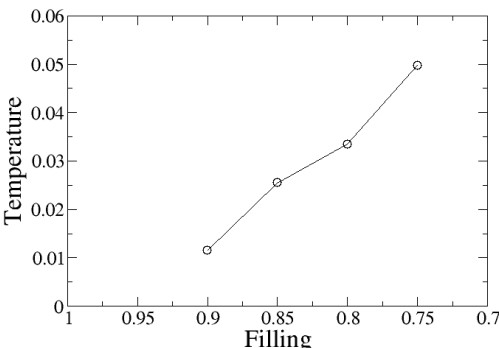

**Figure 4.** The crossover temperature, $T_X$, as a function of filling. The crossover temperature decreases as the filling increases, pointing to the presence of a critical doping.

### 3.2. Spectral Functions

Further insight can be obtained by studying the spectral functions. We calculate the spectral function for different fillings at a low temperature. As QMC solver is used in the DCA calculation, the data for the Green's function are presented in the Matsubara frequency. We use the maximum entropy method to perform the analytic continuation from the imaginary time QMC data into real frequency data [48,49]. Figure 5 shows the spectra of the system at $T = 0.025$, for $N = 0.98$, a filling within the non-Fermi liquid regime. The decrease in density near $\omega = 0$ illustrates the emergence of a pseudogap. Though the simulation time proved too costly to calculate spectra for each simulated filling and temperature, Figure 6 displays the density of states for fillings of $N = 0.80$ and $N = 0.90$ at $T = 0.025$. These densities with a peak close to the Fermi energy show a stark contrast to that of Figure 5, supporting the system being in a Fermi liquid-like state at these parameters.

In order to accurately identify the emergence of the non-Fermi liquid, we also calculate the $Q = 0$ spin susceptibility. The spin susceptibility should increase as temperature decreases until the pseudogap begins to emerge. It then reaches a maximum and begins to decrease with decreasing temperature. Therefore, identifying the temperature where the pseudogap begins to open can be done by accurately capturing the spin susceptibility [30].

However, in this simulation, the spin susceptibility proved to be too noisy, and a clear extraction of its maximum was unreliable. Nonetheless, there was a trend of a decreasing value for the lowest temperatures in all fillings in the non-Fermi liquid regime. Additionally, the temperature associated with the maximum susceptibility appears to decrease as the doping increases.

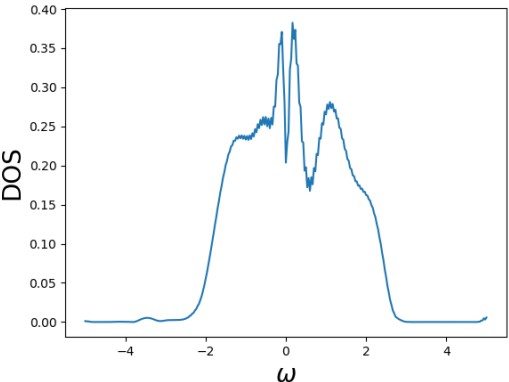

**Figure 5.** Spectra of the system at $T = 0.025$ and $N = 0.98$. The decrease in the density of states near $\omega = 0$ supports our finding that the system is no longer in the Fermi liquid state.

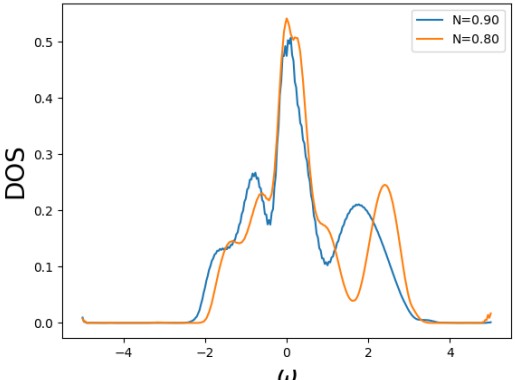

**Figure 6.** Spectra of the model at $T = 0.025$ for $N = 0.80$ and $N = 0.90$. The density of states peak around zero frequency.

## 4. Conclusions

We study the quasi-particle weight of the doped three-dimensional Hubbard model by the dynamical cluster approximation via the continuous time quantum Monte Carlo solver. We find that the quasi-particle weight fits into the crossover form from the Fermi liquid to the marginal Fermi liquid and shows a monotonic decrease in crossover temperature as the filling increases. The putative critical filling nearly coincides with the filling in which the quasi-particle weight decreases sharply as the temperature decreases toward zero. It is tempting to suggest that the marginal Fermi liquid behavior is evidence of a metallic quantum critical point. We emphasize that our conclusion of the existence of the quantum critical point at zero temperature is based on the marginal Fermi liquid at finite temperature.

Metals without broken symmetry are often in the Fermi liquid phase. Any metal that shows a property not consistent with Fermi liquid is interesting physics. The Fermi liquid theory is very robust for metals. It can be understood in terms of the restricted phase space for the low-energy excitation of quasi-particles [29]. It can also be understood as the fixed point of the renormalization group of Fermions with an extended Fermi surface [5]. While the existence of a quantum critical point in the two-dimensional Hubbard model has been proposed and studied, the three-dimensional case has not been thoroughly investigated. A major reason is that these studies are mostly focused on explaining the physics of cuprates,

which are two-dimensional materials. This study fills the void of the important topic of the quantum critical point in the interacting Fermions systems in three dimensions.

The self-energy in the Fermi liquid follows $\omega^2$, so finding the quasi-particle following $\omega$ is often believed to be exotic behavior for metal. We omit the logarithmic correction for the two dimensions. A physical interpretation of the existence of a marginal Fermi liquid is that the phase space argument is not valid. The simplest scenario is that the non-interacting bare density of state has a singularity; however, for the nearest neighbors hopping in a simple cubic lattice, the density of state is finite at all energy. Therefore, a compelling explanation is that a quantum critical point exists, which breaks the phase space argument.

Unfortunately, there is no simple single band model with a concretely demonstrable existence of marginal Fermi liquid. While a few models have been proposed to process such marginal Fermi liquid phase, a recent example is the SYK model [50,51]; however, they are unlikely to be realized in materials. Obtaining numerical evidence to support non-Fermi liquid behavior in a simple and physically realistic model is by itself an interesting and non-trivial finding.

A follow-up point of interest will be to study whether the pseudogap phase in two-dimensional systems, as defined by the partial gap opening or suppression of the density of states, exists for the three-dimensional Hubbard model [30]. We calculate the density of states and show that it is suppressed near the Fermi level; however, we were not able to determine a crossover line between the marginal Fermi liquid and the pseudogap phase due to the noise in the QMC data. Thermodynamic quantities could corroborate our spectral data and possibly support the quantum critical point argument. It has been shown for the two-dimensional model that the entropy peaks at the critical doping [52,53]. A similar effect should be expected in three dimensions. To the best of our knowledge, there is no experimental evidence of a quantum critical point that directly corresponds to the Hubbard model. Even for the two-dimensional model, the putative quantum critical point is likely preempted by the pairing instability. However, there is plenty of evidence of a quantum critical point in Kondo lattice materials [12,13]. It would be worthwhile to study the Kondo lattice model and Periodic Anderson model by the dynamical cluster approximation.

**Author Contributions:** Conceptualization, S.K. and K.-M.T.; Data curation, S.K.; Formal analysis, S.K. and K.-M.T.; Funding acquisition, K.-M.T. and J.M.; Investigation, S.K. and K.-M.T.; Methodology, S.K. and K.-M.T.; Project administration, K.-M.T.; Supervision, K.-M.T. and J.M.; Writing—original draft, S.K. and K.-M.T.; Writing—review & editing, S.K., K.-M.T. and J.M. All authors have read and agreed to the published version of the manuscript.

**Funding:** Samuel Kellar and Ka-Ming Tam are funded by NSF DMR-1728457. Juana Moreno was supported by the U.S. Department of Energy, Office of Science, Office of Basic Energy Sciences under Award No. DE-SC0017861. An award for computer time was provided by the INCITE program. This research also used resources of the Oak Ridge Leadership Computing Facility, which is a DOE Office of Science User Facility supported under Contract DE-AC05-00OR22725.

**Data Availability Statement:** Correspondence and requests for data should be addressed to K.-M.T.

**Acknowledgments:** We thank Mark Jarrell for his many comments and suggestions.

**Conflicts of Interest:** The authors declare no conflict of interest.

## Appendix A

*Appendix A.1. Dynamical Cluster Approximation*

In this appendix, we provide a brief overview of the dynamical cluster approxiamtion (DCA) used in this paper. Additional details can be found in the original paper [38] or in several reviews on quantum cluster methods [39,54,55]. The DCA is an extension of the dynamical mean field theory (DMFT). The DMFT maps the lattice Hubbard model into an effective single-site Anderson impurity problem. The dynamical mean field is given by the non-interacting Green's function of the Anderson impurity model under the assumption that the self-energy for the impurity model is the same as that of the lattice

model $\Sigma(\mathbf{k}, \omega)_{lattice} = \Sigma(\omega)_{impurity}$. The relation between the lattice Green's function and the impurity bare Green's function is given by

$$G_{0,impurity}(\omega) \approx \sum_{\mathbf{k}} \frac{1}{(2\pi)^d} G_{lattice}(\mathbf{k}, \omega) \tag{A1}$$

$$= \sum_{\mathbf{k}} \frac{1}{(2\pi)^d} \frac{1}{G_{0,lattice}^{-1}(\mathbf{k}, \omega) - \Sigma_{lattice}(\omega)}.$$

One can consider that the impurity bare Green's function is the coarse-grained lattice Green's function. Since the coarse graining is performed over the entire first Brillouin zone, the self-energy obtained by solving the impurity model does not have spatial dependence.

The DMFT algorithm starts with a guess of the self-energy. Then the bare impurity Green's function can be obtained via the Dyson equation coarse-grained over the first Brillouin zone. With the bare impurity Green's function, the single impurity Anderson model is solved by numerical solvers such as Quantum Monte Carlo, exact diagonalization, or numerical renormalization group. The impurity solver output is usually given as the impurity Green's function. We can extract the self-energy by inverting the Dyson equation. With the self-energy, the process is then repeated until the self-energy is converged [39,54,55].

The bare lattice Green's function is given by the quadratic part of the Hamiltonian. For example, for the Hubbard model, it is $G_{0,lattice} = \frac{1}{i\omega - \epsilon(\mathbf{k})}$, where $\epsilon(\mathbf{k})$ is the dispersion, for a simple cubic lattice with nearest neighbor hopping only, $\epsilon(\mathbf{k}) = -t[cos(k_x) + cos(k_y) + cos(k_z)] - \mu$. Another example is the periodic Anderson model in which the Hamiltonian is given by

$$H = -t \sum_{<i,j>,\sigma} c_{i\sigma}^\dagger c_{j\sigma} + \mu \sum_{i,\sigma} c_{i,\sigma}^\dagger c_{i,\sigma} + V \sum_{i,\sigma} (c_{i,\sigma}^\dagger f_{i,\sigma} + H.c.) + \tag{A2}$$

$$U \sum_i f_{i\uparrow}^\dagger f_{i\uparrow} f_{i\downarrow}^\dagger f_{i\downarrow} + \epsilon_f \sum_{i,\sigma} f_{i,\sigma}^\dagger f_{i,\sigma},$$

where $c_{i,\sigma}^\dagger$ and $c_{i,\sigma}$ ($f_{i,\sigma}^\dagger$ and $f_{i,\sigma}$) are the creation and annihilation operators for electrons at the conduction (impurity) bands at site $i$ with spin $\sigma$. $t$ is the hopping energy between nearest neighbors, $U$ is the on-site electrons coupling for the electrons in the impurities, and the chemical potential, $\mu$, sets the filling of the conduction band. $V$ is the hybridization, and $\epsilon_f$ is the impurity energy. The formulation of the DCA is the same as that of the Hubbard model, except that the bare lattice Green's function is given as $G_{0,lattice}(\mathbf{k}, i\omega) = (i\omega - \epsilon_f - \frac{V^2}{i\omega - \epsilon(\mathbf{k})})^{-1}$.

A key idea of the DCA is to partially restore the spatial dependence by introducing an impurity cluster. The momentum dependence is obtained by coarse graining within the cluster of $N_c$ impurities. The impurities span the first Brillouin zone, which is divided into $N_c$ equal size patches. The momentum at the center of each patch is labeled as $\mathbf{K}$. Unlike the DMFT, the self-energy gains momentum dependence in the DCA. The lattice self-energy is approximated by $\Sigma(\mathbf{k} + \mathbf{K}, \omega)_{lattice} \approx \Sigma(\mathbf{K}, \omega)_{cluster}$. The relation between the lattice Green's function and the cluster bare Green's function is given by

$$G_{0,cluster}(\mathbf{K}, \omega) \approx \sum_{\mathbf{k}} \frac{1}{(2\pi)^d} G_{lattice}(\mathbf{k} + \mathbf{K}, \omega) \tag{A3}$$

$$= \sum_{\mathbf{k}} \frac{1}{(2\pi)^d} \frac{1}{G_{0,lattice}^{-1}(\mathbf{k} + \mathbf{K}, \omega) - \Sigma_{lattice}(\mathbf{K}, \omega)},$$

where the summation over $\mathbf{k}$ is restricted to the patch corresponding to the impurity site with momentum $\mathbf{K}$.

*Appendix A.2. Minus Sign of the Quantum Monte Carlo Impurity Solver*

In this work, we obtain the momentum-dependent self-energy $\Sigma(\mathbf{K}, \omega)$ by solving the impurity cluster problem by the continuous time Quantum Monte Carlo (CTQMC). The CTQMC is based on expanding the interaction term with respect to the non-interacting limit. The method in general does not guarantee all the configurations have a positive weight. It is interesting and prominently important to monitor the so-called average sign in the Monte Carlo sampling. If the average sign is close to zero, meaningful statistics may not be attainable in a practical amount of computer time.

A parameter, usually denoted as $\alpha$, shifts the Hamiltonian by a constant in the CTQMC [41,42]. In principle, this parameter can be tuned to slightly adjust the sign at the expense of decreasing the sampling of higher-order terms in the expansion. The average sign as a function of temperature and filling factor $N$ is shown in Figure A1. It is clear that the sign behaves differently in the non-Fermi liquid region, making the simulation for fillings larger than 0.95 extremely time-consuming and difficult.

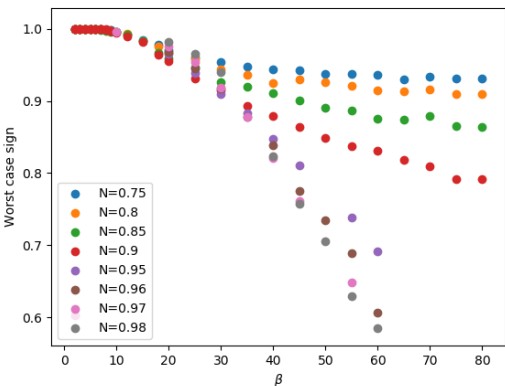

**Figure A1.** Average sign as a function of inverse temperature, $\beta = 1/T$, for different fillings. We set $\alpha = 0.51$ for all the simulations [41,42].

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
