# Peer review of "Non-Fermi Liquid Behavior in the Three-Dimensional Hubbard Model"

_crystals, doi:10.3390/cryst13010106_

Round 1

Reviewer 1 Report

In this paper, the authors introduced a numerical study on the non-Fermi liquid behavior of a three dimensional strongly correlated system. The Hubbard model in a cubic lattice is simulated by the dynamical cluster approximation, in particular the quasi-particle weight is calculated at finite dopings for range of temperatures. At a finite doping near the putative quantum critical point, we find evidence of a separatrix between Fermi liquid and non-Fermi liquid regions as the doping increases. Our results suggest that a marginal Fermi liquid and possibly a quantum critical point exist in three dimensional interacting Fermi systems. The idea behind this is interesting. However, I still have quite a number of concerns in this manuscript. There are times where there are not enough data to support the conclusions of the author. Please see some of the major concerns below.

1.The information for the example of the tiling used for the sixteen site cluster chosen for our simulation of the dynamical cluster approximation is not enough more parameters need to be added (x,y,z) grid size not clear and etc. The authors should give much more information about this. So the readers can get its reproducibility. 

2.  The authors should give much more information about the novelty of this paper, especially the effect of using this new study on the non-Fermi liquid, which applications can be used this method?

3. More references need to be included in the introduction part to understand the applications of using doping for device applications.

a.     Optimizations of Si PIN diode phase-shifter for controlling MZM quadrature bias point using SOI rib waveguide technology

Optics & Laser Technology, 2021

b.     Optimizations of thermo-optic phase shifter heaters using doped silicon heaters in Rib waveguide structure

Photonics and Nanostructures-Fundamentals and …,

4.  Much more discussion about the results should be given in this paper, especially the author needs to provide enough physicals mechanism analysis about the results.

Reviewer 2 Report

Please see the file blew.

Reviewer 3 Report

\documentstyle[12pt]{article}
\textwidth 160 mm
\textheight 225 mm
\topmargin -25mm
\hoffset -1cm
\pagestyle{empty}
%\usepackage{color}

\begin{document}

\begin{center}
{\Large\bf {Referee report \\to the paper crystals-2078408 entitled\\
{\it "\textit{Non-Fermi Liquid Behavior in the Three Dimensional Hubbard Model}" \\
by Samuel Kellar, Ka-Ming Tam and Juana Moreno}}}
\end{center}

The authors of the paper crystals-2078408
have numerically studied the non-Fermi liquid behavior of a three dimensional
strongly correlated system.
The Hubbard model in a cubic lattice is simulated by the dynamical
cluster approximation, in particular, the quasi-particle weight is calculated at finite dopings for a
range of temperatures. The quasi-particle weight of the doped three-dimensional Hubbard model is investigated by
the dynamical cluster approximation via the continuous-time quantum Monte Carlo solver.
It is found that the quasiparticle weight fits into the crossover form from the Fermi liquid to
the marginal Fermi liquid, and shows a monotonic decrease of crossover temperature as
the filling increases. The putative critical filling nearly coincides with the filling in which
the quasi-particle weight decreases sharply as the temperature decreases toward zero. It
is tempting to suggest that the marginal Fermi liquid behavior is evidence of a metallic
quantum critical point.

At finite doping near the putative quantum critical point, it is found evidence
of a separatrix between Fermi liquid and non-Fermi liquid regions as the doping increases. The authors claim that the obtained  
results suggest that a marginal Fermi liquid and possibly a quantum critical point exist in three-dimensional interacting Fermi systems.

I have the following comments and suggestions:

i) The abstract of the paper has not been well written—the current form of the abstract looks like an introduction to the topic.
It would be better if the author could slightly revise the text of the abstract in a form presenting the main results from the
the main part of the paper.

ii) The Conclusions section is concise. I advise the author to summarize in detail the obtained results
with more detailed scientific discussions.

iii) The extended plots and figures are requested to be produced in order to illustrate the results.

iv) More detailed analysis of the scientific research issues has to be performed toward
the detailed description of the obtained results. At the moment it looks like standard calculations
with the figures 1-5 produced.

v) Since the Crystals journal is oriented toward publication of the papers related to finding original solutions and results the author has to explore
possible inclusion of more actual detailed results.

I do not think that the performed research of standard nature  with some results
would warrant a
publication in a high impact journal like the \textbf{Crystals} specialized in publishing original results
having mathematical and physical importance. At the moment
it really contains little original scientific results.
What is the advantage of this paper from the scientific point of view?
I do not think that the submission is suitable for publication in the \textbf{Crystals} journal.

\end{document}

Round 2

Reviewer 1 Report

The new version it is much better so paper can be published after adding the recommend Ref was suggsted such as:

Optimizations of Si PIN diode phase-shifter for controlling MZM quadrature bias point using SOI rib waveguide technology

Author Response

We thank the referee for the recommendation of publication.

As we explained in the reply to the last round of review, far as we understand, there are no immediate applications of the marginal Fermi liquid in the context of waveguide technology. We have added additional references (ref. 18, 19, 27, and 28) for the experiments, theory, and review of marginal Fermi liquid in the revised manuscript.

Reviewer 3 Report

Comments on  Revised Manuscript crystals-2078408   "Non-Fermi Liquid Behavior in the Three Dimensional Hubbard Model" by authors Ka-Ming Tam,  Juana Moreno, Samuel Kellar     The authors in the revised version have addressed my comments. The paper in the current form is suitable to be accepted for publication in Crystal journal by MDPI. 

Author Response

We thank the referee for the recommendation of publication.